Chargaff’s second parity rule lies at the origin of additive genetic interactions in quantitative traits to make omnigenic selection possible

Matkarimov Bakhyt T. bmatkarimov@nu.edu.kz bakhyt.matkarimov@gmail.com 1 2
Saparbaev Murat K. murat.saparbaev@gustaveroussy.fr 3 4
1 National Laboratory Astana, Nazarbayev University , Astana , Kazakhstan
2 L.N.Gumilev Eurasian National University , Astana , Kazakhstan
3 Groupe «Mechanisms of DNA Repair and Carcinogenesis», CNRS UMR9019, Gustave Roussy Cancer Campus, Université Paris-Saclay , Villejuif , France
4 Al-Farabi Kazakh National University , Almaty , Kazakhstan
Gillespie Joseph
Electronic publication date: 2023 Dec 14
Publication date: 2023
Volume: 11
Electronic Location ID: e16671
Received 2023 Jul 21; Accepted 2023 Nov 22
Copyright: ©2023 Matkarimov and Saparbaev
Copyright year: 2023
Copyright holder: Matkarimov and Saparbaev
License: This is an open access article distributed under the terms of the Creative Commons Attribution License, which permits unrestricted use, distribution, reproduction and adaptation in any medium and for any purpose provided that it is properly attributed. For attribution, the original author(s), title, publication source (PeerJ) and either DOI or URL of the article must be cited.
License URL: https://creativecommons.org/licenses/by/4.0/

Keywords: Intra-strand DNA symmetry, Single nucleotide polymorphisms, Quantitative trait, Infinitesimal model, Statistical entanglement, Integral characteristics, Chargaff’s second parity rule, Nucleotide composition bias, Random mutations, Continous trait variation

Funding: The Science Committee of the Ministry of Education and Science of the Republic of Kazakhstan AP09260233, program BR11765589 The French National Research Agency ANR-22 CE1212-0034-01 Electricité de France RB 2020-02 and RB 2021-05 This research was funded by the Science Committee of the Ministry of Education and Science of the Republic of Kazakhstan (grant AP09260233, program BR11765589) to Bakhyt T. Matkarimov; and by the French National Research Agency (grant ANR-22 CE1212-0034-01) and Electricité de France (grants RB 2020-02 and RB 2021-05) to Murat K. Saparbaev. The funders had no role in study design, data collection and analysis, decision to publish, or preparation of the manuscript.

==============================
Background

Francis Crick’s central dogma provides a residue-by-residue mechanistic explanation of the flow of genetic information in living systems. However, this principle may not be sufficient for explaining how random mutations cause continuous variation of quantitative highly polygenic complex traits. Chargaff’s second parity rule (CSPR), also referred to as intrastrand DNA symmetry, defined as near-exact equalities G ≈ C and A ≈ T within a single DNA strand, is a statistical property of cellular genomes. The phenomenon of intrastrand DNA symmetry was discovered more than 50 years ago; at present, it remains unclear what its biological role is, what the mechanisms are that force cellular genomes to comply strictly with CSPR, and why genomes of certain noncellular organisms have broken intrastrand DNA symmetry. The present work is aimed at studying a possible link between intrastrand DNA symmetry and the origin of genetic interactions in quantitative traits.

Methods

Computational analysis of single-nucleotide polymorphisms in human and mouse populations and of nucleotide composition biases at different codon positions in bacterial and human proteomes.

Results

The analysis of mutation spectra inferred from single-nucleotide polymorphisms observed in murine and human populations revealed near-exact equalities of numbers of reverse complementary mutations, indicating that random genetic variations obey CSPR. Furthermore, nucleotide compositions of coding sequences proved to be statistically interwoven via CSPR because pyrimidine bias at the 3rd codon position compensates purine bias at the 1st and 2nd positions.

Conclusions

According to Fisher’s infinitesimal model, we propose that accumulation of reverse complementary mutations results in a continuous phenotypic variation due to small additive effects of statistically interwoven genetic variations. Therefore, additive genetic interactions can be inferred as a statistical entanglement of nucleotide compositions of separate genetic loci. CSPR challenges the neutral theory of molecular evolution—because all random mutations participate in variation of a trait—and provides an alternative solution to Haldane’s dilemma by making a gene function diffuse. We propose that CSPR is symmetry of Fisher’s infinitesimal model and that genetic information can be transferred in an implicit contactless manner.

Introduction

Similar to Mendel’s laws of heredity, Chargaff’s second parity rule (CSPR), also referred to as intrastrand DNA symmetry, was accidentally determined by measurement of nucleotide composition of single-stranded genomic DNAs (Rudner, Karkas & Chargaff, 1969) and has later been rediscovered and confirmed by others (Baisnée, Hampson & Baldi, 2002; Forsdyke, 1995a; Lobry, 1995; Prabhu, 1993; Qi & Cuticchia, 2001; Sueoka, 1995). Portions of this text were previously published as part of a preprint (Matkarimov & Saparbaev, 2023). It is noteworthy that intrastrand DNA symmetry can be extrapolated to any pair of reverse complementary short oligonucleotides for a sufficiently long DNA sequence (>100 kbp); this means near-exact equalities of reverse complementary pairs of mono- and oligo-nucleotides in one strand of cellular DNA (Albrecht-Buehler, 2006; Fickett, Torney & Wolf, 1992; Prabhu, 1993). CSPR holds for genomes of cellular organisms (including eukaryotic, bacterial, and archaeal chromosomes) and double-stranded DNA viruses, whereas organellar genomes (mitochondria and plastids) smaller than ∼20–30 kbp, single-stranded viral DNA genomes, and any type of RNA genome do not obey CSPR (Mitchell & Bridge, 2006). Intriguingly, intrastrand symmetry is maintained in genomic DNA of all cellular organisms despite billions of years of divergent evolution of various species from one common ancestor. At the scale of an autonomously segregating replicon such as a chromosome or entire genome, cellular organisms obey CSPR with small deviations (less than 1%) from the mononucleotide parity rule (Mitchell & Bridge, 2006). It remains unknown what the mechanisms are that force cellular genomes to adopt CSPR and why genomes of certain noncellular organisms have broken intrastrand DNA symmetry. Sueoka showed that parity rule 2 (PR2, here referred to as CSPR) theoretically holds under no-strand-bias conditions, where mutation rates are similar between the two strands in a DNA duplex (Sueoka, 1995). However, on the local scale, PR2 or CSPR is often violated because the mutation/selection rates in the two strands of the DNA molecule are not at equilibrium (Lobry, 1999). This disparity arises mainly due to intrinsic asymmetry in DNA replication and transcription and directional mutation pressure. Nevertheless, in sufficiently long DNA segments, the parity is always restored due to alternation of DNA sequences with different signs of bias between Watson and Crick strands (Rapoport & Trifonov, 2013). Several mechanisms have been proposed to explain the origin of CSPR: formation of stem-loop structures (Forsdyke, 1995b), a combination of a strand inversion and duplication (Albrecht-Buehler, 2006), CSPR’s being a feature of the primordial genome (Zhang & Huang, 2008), DNA free energy equilibrium (Fariselli et al., 2021), and mutation rate interrelations (Pflughaupt & Sahakyan, 2023). Previously, we introduced the concept of DNA strand equivalence and suggested that the broken intrastrand symmetry of vertebrate mitochondrial genomes is due to asymmetric DNA strand inheritance (Matkarimov & Saparbaev, 2020).

Here, we sought to examine a possible biological role of the evolutionarily conserved intrastrand DNA symmetry in cellular organisms by exploring the link between CSPR and continuous phenotypic variation of complex traits of higher eukaryotes. Quantitative or complex traits of plants and animals are characterized by measurable phenotypic variation within a population owing to underlying genetic variation and an environmental influence. Complex traits such as human height do not follow simple Mendelian inheritance patterns because their phenotypic variations cannot be explained by the segregation of a single genetic factor. In 1919, Ronald Fisher showed that (i) the variation of continuous traits could be explained by Mendelian inheritance laws if an infinite number of genetic loci control the trait and (ii) each locus contributes additively to the trait (Fisher, 1919). In his work, Fisher introduced the infinitesimal model, also known as the polygenic model, based on the idea that variation of a quantitative trait is influenced by an infinitely large number of genes, each of which with an infinitely small (infinitesimal) effect on the phenotype. Importantly, quantitative traits are far more common as compared to simple Mendelian traits; moreover, even monogenic traits can exhibit phenotypic heterogeneity among organisms of the same genotype because of complex genetic interactions such as epistasis, incomplete penetrance, variable expressivity, and pleiotropy (Domingo, Baeza-Centurion & Lehner, 2019). Given that quantitative traits can be extremely polygenic, they are examined using statistical techniques such as genome-wide association studies (GWASs) rather than classic molecular biological methods (Griffiths et al., 2015). The infinitesimal model provides a basis for statistical analysis and selection theory, which are used in plant and animal breeding programs with great success. Nevertheless, the model remains a theoretical abstraction, because in reality, all living organisms have finite-size genomes each containing a limited number of genes. Prior to the era of next-generation DNA sequencing, it had been impossible to verify the infinitesimal model for a quantitative trait in a genetically heterogeneous population. Variants discovered through GWAS have accounted for only a small percentage of predicted heritability, and this has led to the missing heritability problem (Manolio et al., 2009). With advances in DNA genotyping technologies, the problem has been in part resolved via evidence that most of the heritability could be accounted for by common single-nucleotide polymorphisms (SNPs) missed by GWASs because their effect sizes have fallen below genome-wide statistically significant thresholds (Claussnitzer et al., 2020; Purcell et al., 2009; Yang et al., 2010). Furthermore, it has been demonstrated that for the most complex traits, the majority of the SNP-associated heritability is distributed uniformly across the genome (Loh et al., 2015; O’Connor et al., 2019; Shi, Kichaev & Pasaniuc, 2016). Recent studies estimated that 10,000 to 100,000 causal variants contribute to a quantitative trait in general, in remarkable agreement with the infinitesimal model (O’Connor et al., 2019; Zhang et al., 2018).

Although many common variants involved in quantitative traits have been identified, the molecular mechanisms through which they contribute to phenotypic variation remain poorly understood (Albert & Kruglyak, 2015). For example, the infinitesimal model postulates that each genetic locus contributes additively to a trait, while it remains unclear how genetic interactions can act in the additive manner and what the molecular mechanisms of this complementarity are. Boyle and colleagues when analyzing GWAS datasets of complex traits including human height, schizophrenia, rheumatoid arthritis, and Crohn’s disease found that the SNPs associated with the traits were evenly distributed all over the genome rather than clustered within trait-specific genes (Boyle, Li & Pritchard, 2017). To explain these observations, they proposed the concepts of “core” genes (a limited number of disease-specific genes with established biological roles) and an “omnigenic” model, in which the majority of active genes, rather than a specific set of core genes, can affect every complex trait. For example, the total set of expressed genes in an affected tissue or a cell in general outnumbers “core” genes by 100:1 or more; consequently, the sum of small effects across peripheral genes can far exceed the genetic contribution of variants of core genes because of the highly interconnected cell regulatory networks (Boyle, Li & Pritchard, 2017; Sinnott-Armstrong et al., 2021). In support of this notion, GWAS signals for quantitative traits and disease-associated SNPs have been found to be enriched within predicted gene-regulatory elements and within open chromatin that is active in cell types relevant to a disease, respectively (Farh et al., 2015; Finucane et al., 2015; Kundaje et al., 2015; Wood et al., 2014).

The highly polygenic character of quantitative traits and common diseases may also be explained by negative selection toward large-effect mutations in critical or “core” genes and loci that perform biologically important functions. In support of this idea, statistical modeling of 33 complex traits revealed that heritability is distributed more evenly among common variants than among specific SNPs in functionally important regions owing to selective constraints (O’Connor et al., 2019). Of note, this phenomenon also lies at the origin of Haldane’s dilemma (Haldane, 1957), in which large-effect-size mutations lead to high reproductive cost. Researchers have been able to identify many genetic factors, such as constitutively active peripheral genes, involved in complex traits, but determining the specific molecular mechanisms through which these factors contribute to phenotypic variation is becoming more and more challenging without clear insight into the whole picture (Connally et al., 2022; Sinnott-Armstrong et al., 2021). Here, we attempt to identify the nature of genetic interactions involved in complex traits by means of a nonreductionist approach.

Fisher’s infinitesimal model and the “omnigenic” hypothesis explain genetics of highly polygenic complex traits by postulating the existence of specific interactions between multiple loci in a genome. Although both models imply that these interactions are due to highly interconnected genetic and cellular networks, it remains unclear: (i) why interactions are additive; and (ii) what the molecular mechanisms are that link common variants to the variation of polygenic traits. Here we investigate how CSPR interferes with the transmission of genetic information from common variants to polygenic traits and from somatic mutations to cancer. We propose below that in cellular organisms, CSPR serves as a key for decoding the information present in a genome into a phenotype.

Materials & Methods

Analysis of mutation spectra derived from SNP variation in natural populations

NCBI dbSNP is a public collection of nucleotide variations for different species from a wide variety of sources (Sherry et al., 2001). DbSNP does not track individual samples and represents all identified genetic variation aligned to a reference genome. SNP/indel statistics were computed by means of NCBI dbSNP dataset build 142 (October 14, 2014) with reference genomes: human GRCh38 build 38.1 and mouse GRCm38.p2 build 38.3. Every SNP was counted once from the top strand, only validated SNP/indels were counted, and SNPs from sex chromosomes were excluded. Every SNP/indel was supplemented with left/right flanking nucleotide sequences. There were 192 directed SNPs of type NXN↔NYN, where X/Y are ref/alt nucleotides. These 192 SNPs were grouped by 88 symmetric pairs based on ref/alt interchange, e.g., the count of group AGC↔ATC was compared with the count of reverse complementary group GCT↔GAT. In addition, there were eight self-complementary SNPs, which are presented as a single group (e.g., AGT↔ACT, CAG↔CTG, …). For more details, see Table 1 and Tables S1 and S2.

Table 1 Mutation spectra inferred from single nucleotide polymorphisms in human and mouse populations.a

No	N X N	N Y N	X	Y	Total number of SNPs	Fraction	N′X′N′	N′Y′N′	X′	Y′	Total number of SNPs	Fraction	Difference (%)	
	Human SNPs	
1	AGC	ATC	G	T	312,720	0.37	GAT	GCT	A	C	310,244	0.36	0.79	
2	CAA	CTA	A	T	227,826	0.27	TAG	TTG	A	T	229,630	0.27	0.79	
3	ACT	ATT	C	T	2367,625	2.77	AAT	AGT	A	G	2375,298	2.78	0.32	
4	AAA	AGA	A	G	1729,945	2.02	TCT	TTT	C	T	1734,336	2.03	0.25	
5	TAT	TTT	A	T	544,308	0.64	AAA	ATA	A	T	543,030	0.64	0.23	
6	GAG	GTG	A	T	287,742	0.34	CAC	CTC	A	T	288,408	0.34	0.23	
7	GCG	GTG	C	T	2008,008	2.35	CAC	CGC	A	G	2008,195	2.35	0.01	
8	TAA	TGA	A	G	1448,254	1.69	TCA	TTA	C	T	1448,196	1.69	0.00	
	Mouse SNPs	
1	CGA	CTA	G	T	38,698	0.26	TAG	TCG	A	C	38,235	0.25	1.20	
2	AAG	ATG	A	T	88,927	0.59	CAT	CTT	A	T	89,884	0.59	1.06	
3	GAG	GCG	A	C	50,929	0.34	CGC	CTC	G	T	50,724	0.33	0.40	
4	CAG	CCG	A	C	56,636	0.37	CGG	CTG	G	T	56,410	0.37	0.40	
5	GAT	GTT	A	T	63,675	0.42	AAC	ATC	A	T	63,432	0.42	0.38	
6	ACT	ATT	C	T	384,248	2.53	AAT	AGT	A	G	384,516	2.54	0.07	
7	TAT	TTT	A	T	103,335	0.68	AAA	ATA	A	T	103,323	0.68	0.01	
8	AAG	AGG	A	G	299,963	1.98	CCT	CTT	C	T	299,979	1.98	0.01	
Notes.

a Data are shown only for eight reverse complementary pairs of triplets out of total 48 pairs. For the complete data set see Table S1.

Analysis of the nucleotide composition at different codon positions in bacterial and human protein-coding sequences (CDSs)

CDSs were selected from curated databases: CCDS for humans/mice (Pujar et al., 2018) and NCBI RefSeq for bacterial species (O’Leary et al., 2016). For bacterial genomes, we excluded CDSs of hypothetical proteins. For every selected CDS, we counted statistics of each single nucleotide for the 1st, 2nd, and 3rd codon positions and CDS total statistics and then computed corresponding A/T and G/C ratios; it follows that we have four (A/T, G/C) ratios that include the ratios for three different positions plus the ratio for all three positions together. For human CDSs, we created a two-dimensional plot of (A/T, G/C) for every CDS separately, in total 33,420 CDSs, with different colors for codon positions, and the black color for the CDS total. For bacteria, we computed accumulative statistics; e.g., single-nucleotide counts are the sum of all CDSs from a single bacterial species, and this produces four (A/T, G/C) ratios for each species separately, in total 4,038 species having 60,417 different names of proteins (Table S3).

Analysis of the nucleotide composition at different codon positions in CDSs of humans and mice by taking in account orientation of the genes relative to a reference strand

We processed the CDS data from the curated database: CCDS for humans/mice (Pujar et al., 2018) (Table S4). We computed statistics/counts for nucleotide sequences of lengths of 1 (mononucleotides), 2 (dinucleotides) and 3 (trinucleotides) starting at the 1st, 2nd, and 3rd codon positions for two cases: for CDS sequences proper (a strand-independent case) and for the CDS sequences read from a single strand (a strand-dependent case), e.g., reverse complementary CDSs for the set of proteins located on a different strand. For the reverse complementary CDSs, we set up the 1st, 2nd, and 3rd codon positions relative to the 5′ → 3′ DNA reverse-stand orientation. Additionally, we computed the total count as the sum of counts for all codon positions. To compare statistical counts of nucleotide sequences and their reverse complementary counterparts, we used the percentage difference formula 200  × (A − B)/(A + B). In a single row of the supplementary data files, we show the name of a sequence (SEQ); its counts at the first (P1), second (P2), third (P3) codon positions; and the total count (TOTAL). Then, we show the same data for the reverse complementary sequence; after that, the percentage difference for 1st, 2nd, and 3rd codon positions and the total between the sequence and it reverse complementary sequence (for more details see Table S5).

Results and Discussion

Continuous variation of complex traits enables natural selection

Because of their polygenic multilocus nature, quantitative traits exhibit a continuous normal distribution of phenotypic variations. According to the infinitesimal model introduced by Fisher, continuous phenotypic variation of a quantitative trait influenced by an infinitely large number of genetic loci in a population has a normal distribution (Fig. 1A) (Nelson, Pettersson & Carlborg, 2013).

Figure 1 The normal distribution of values of a quantitative trait in a population echoes selective pressure exerted on each individual.

(A) Stabilizing selection favors average individuals in a population. (B) Directional selection shifts the distribution of allele frequencies or genetic variations in a population thereby adjusting trait values to environmental changes. Pink arrows denote selective pressure; the blue arrow indicates directional selection.

Here we propose that this bell-shaped distribution also reflects differential selective pressure or fitness of each individual in the population. For example, if human height is considered, individuals with close-to-average heights experience less selective pressure and are present in greater numbers in the population because they are better adapted, whereas the shortest and tallest individuals endure high selective pressure and thus are present in much lower numbers. Similarly, in a multicellular organism, a population of somatic cells with mean variations of a trait undergoes less selective pressure and is present in greater cell numbers in a given tissue or organ because these cells are better adapted to an internal body environment, whereas cells with extreme phenotypes undergo high selective pressure and are present at much lower frequencies because they are less fit. However, as shown in Fig. 1B, variations in the external environment and organism homeostasis can change the direction of selection pressure, and as a consequence, organisms and somatic cells, respectively, exhibiting positive (+) trait variation may have an advantage; this may shift common variants or the allele frequency distribution to the right of the horizontal axis.

The cell as a part of a highly complex multicellular organism experiences constant selective pressure because of the internal body environment, which imposes severe restraints on somatic cells. Multiple factors, such as limited availability of oxygen and nutrients, immune-system surveillance, cell–cell contacts, and cell-signaling factors (among many others) that tightly control growth and proliferation of individual cells within the body, all together create selective pressure on a population of somatic cells to integrate every part and function of the body as a coherent whole. In fact, cell differentiation—in which changes in gene transcription programs are mediated by epigenetic mechanisms without alterations in the primary DNA sequence—can be regarded as a way for a somatic cell to adapt to the continuously changing inner body environment in complex multicellular organisms during their development and aging. On the other hand, random somatic mutations that inevitably accumulate after each cell division are subject to strong negative selection pressure that eliminates cells with deleterious and oncogenic mutations. Somatic mutagenesis is a random stochastic process, and the majority of mutations observed in higher eukaryotes occur in noncoding regions of the genome and thus may evade the strong selective pressure. It is widely accepted that in cancer, only a few (driver) mutations strongly influence its progression, with the remaining (passenger) mutations being neutral or slightly deleterious (Vogelstein & Kinzler, 2015). The recent Pan-Cancer Analysis of Whole Genomes (PCAWG) variant dataset of 43,778,859 single-nucleotide variants revealed only thousands of mutations that can be identified as drivers, with the mean of five driver mutations per tumor, whereas the remaining 99.99% of single-nucleotide variants have been referred to as passenger variants, with no known function and effect on fitness (ICGC/TCGA Pan-Cancer Analysis of Whole Genomes Consortium, 2020). Nevertheless, another analysis of the PCAWG variant dataset suggests that the combined effect of passenger mutations together with undetected weak drivers can provide an additional 12% effect for identifying cancer phenotypes, beyond driver mutations (Kumar et al., 2020).

Despite huge technological progress in DNA sequencing and analysis of big data, fundamental gaps remain in our understanding of how normal cells evolve into tumor cells. It is assumed that the absolute majority of somatic and passenger mutations in normal and cancer cells, respectively, are either neutral or slightly deleterious. However, this assumption is challenged by several important observations. First, the existence of genetic interaction networks between multiple genes (on a genome-wide scale) that can determine phenotypic variation of the observable traits including the expression of monogenic traits such as rare highly penetrant deleterious mutations of genes. It was shown that common genetic variants can critically contribute to incomplete penetrance of severe Mendelian diseases, and that not all individuals with a given genotype (mutant allele) display the corresponding phenotype (Chen et al., 2016a). This phenomenon was observed also in research articles on other model organisms (reviewed in ref. Goldstein & Ehrenreich, 2021). Second, numerous studies have convincingly demonstrated that calorie restriction (CR), as compared to a control ad libitum (AL) regimen, dramatically delays spontaneous tumor development in laboratory animals (Chen et al., 2016b). A change in a dietary regimen influences the body’s homeostasis in a way that may reduce the fitness of cancer-prone cells as compared to normal healthy cells and may prevent propagation of the former in tissues. Here, we propose that accumulation of a sufficient number of random mutations in somatic cells inevitably results in phenotypic variations owing to small additive genetic interactions. Trait variations in somatic cells such as glucose uptake and the proliferation rate are subject to natural selection. Therefore, the CR regimen should favor the growth of cells with a lower proliferation rate and glucose consumption, whereas precancerous cells are expected to be under increased negative selection pressure, and this may prevent their propagation. On the basis of these observations, we believe that common variants and passenger mutations can influence phenotypic expression of a simple Mendelian trait and cancer-driver mutation, respectively, by uncovering complex highly interconnected gene regulatory networks. Below, we analyze the role of intrastrand DNA symmetry in random mutations and phenotypic diversity.

Intrastrand DNA symmetry and genetic variations in cellular organisms

CSPR can be interpreted as the law of large numbers postulating that as the number of identically distributed randomly generated variables increases, their sample mean approaches their distribution mean. Therefore, intrastrand DNA symmetry can be considered a general statistical property of cellular genomes composed of independently segregating replicons, which is maintained over evolutionary history of life. This means that a sufficiently large number of genetic variations in a population have a statistical tendency: every single random mutation must be “compensated” by another inverse complementary mutation elsewhere in the genome to maintain the “symmetry.” This mechanism of compensation manifests itself through the assembly of reverse complementary mutations into pairs occurring in one DNA strand (e.g., probability of the C→T mutation is nearly equal to that of G→A); this effect allows to preserve intrastrand DNA symmetry within an entire segregating replicon(s). As shown in Table 1, our analysis of the patterns of spontaneous mutations in the trinucleotide sequence context inferred from SNPs observed in human and mouse populations revealed that in a single DNA strand of human and murine genomes, the number of X↔Y variations in triplet NXN (referred to here as —NXN↔NYN—) is nearly equal to the number of X’↔Y’ variations occurring in reverse complementary triplet N’X’N’ on the same DNA strand (referred to here as —N’X’N’↔N’Y’N’—, where, —— stands for the total number of instances of a variation, N is any nucleotide, X and Y are different versions of a nucleotide, and N’, X’, and Y’ are the corresponding reverse complementary counterparts).

For example, the number of SNPs AGT↔AAT, referred to here as —AGT↔AAT—, observed in the human population, is 2,375,298 (100%), which is nearly equal to the number of SNPs representing its reverse complementary version —ACT↔ATT—, which is observed 2,367,625 times (99.7%) in the same DNA strand. Similar statistical parities were observed in mouse populations there: —AGT↔AAT— ≈ —ACT↔ATT— [384,516 (100%) and 384,248 (99.99%) times, respectively] with less than a 1% difference (Table 1 and Table S1). Furthermore, our analysis of small insertion/deletion (indel) polymorphism in the human population revealed near-exact equalities in the number of indels within 18 out of 32 reverse complementary pairs of trinucleotides (Table S2). It should be noted that these 18 trinucleotides have a sequence context in which a base corresponding to an indel is different from the neighboring bases (i.e., —TΔC↔TGC— ≈ —GΔA↔GCA—). By contrast, for the remaining 14 out of the 32 pairs, the correct pairwise alignment is simply impossible because of their sequence contexts’ particularity in which a base corresponding to an indel is the same as a neighboring base (e.g., —TΔG↔TTG— ≠ —CΔA↔CAA—). These observations indicate that during evolution, in a population of cellular organisms, the accumulation of random mutations represented here in the form of SNPs and indels is a necessary and sufficient condition for driving chromosomal DNA to the state of intrastrand symmetry equilibrium and for maintaining it. Thus, stochastic processes such as random mutations can lead genome evolution toward the symmetry of nucleotide compositions, and this phenomenon should be regarded as one of major forces of evolution, which (as readers will see below) provides a great selective advantage to cellular organisms.

Statistical interdependence of the nucleotide compositions is revealed by a specific bias at the 3rd position of the codon

The analysis of mutation spectra inferred from SNPs suggested that CSPR is an essential property of cellular organisms, which restrains the patterns of random genetic mutations accumulating them in the pairwise statistically interwoven manner. Local deviations from intrastrand DNA symmetry in the majority of coding DNA sequences (CDSs), referred to as Szybalski’s rule (Szybalski, Kubinski & Sheldrick, 1966) (when the number of purines exceeds that of pyrimidines), suggest that at the gene level, directional mutational pressure and selective constraints counteract the homogenizing force of CSPR. To examine this closely, we measured deviations from intrastrand DNA symmetry of the nucleotide composition at each of the three positions of a codon in bacterial and human CDSs. In Fig. 2, each codon position is plotted according to its purine/pyrimidine (Pur/Pyr or G/C and A/T) content [note that CDSs are analyzed regardless of whether they are on the top sense (+) Watson strand or bottom antisense (−) Crick DNA strand of a genomic sequence].

Figure 2 Nucleotide compositions of different positions of a codon in bacterial and human proteomes.

G/C and A/T ratios for the 1st position of a codon is denoted as a red square, the 2nd position is green, the 3rd position is blue, and all three positions together are presented as a black square. (A) Bacterial ORFs. Each point corresponds to a proteome of one bacterial strain. (B) Human ORFs. Each point corresponds to a single ORF. (C, D) Same as panels A and B, respectively, but contain isochores, the dashed lines drawn through 51–53% of all points having values close to average, in order to make the data on graph accessible to readers with color-blindness.

When all three positions of a codon are taken together, the Pur/Pyr ratio of nucleotide composition of each triplet (denoted as black squares) in bacterial proteomes exhibits relatively weak deviations from CSPR, with G/C and A/T contents nearly equal to one (≈ 1), as compared to that of single codon positions (denoted as colored squares) (Figs. 2A, 2C). Remarkably, in an absolute majority of bacteria, the 1st position of a codon (red square) has a highly increased purine content (Pur/Pyr ratio ≥ 1.5), while positions 2 and 3 (green and blue squares, respectively) contain an increased C and T content (G/C and A/T ratios ≤ 1), respectively, indicating that the nucleotide composition at the 2nd and 3rd positions of a codon compensates the very strong bias of the purine content at the 1st position of a codon. As a result, when all three positions of a codon are taken into account, more symmetry in the nucleotide compositions of triplets can be seen, which in turn dramatically reduces the deviations of CDSs from CSPR at the gene level in bacteria. In the human proteome, we also observed the phenomenon of compensation of the purine bias in the nucleotide composition at the 1st position of a codon by that of 2nd and 3rd positions, resulting in a Pur/Pyr ratio of ∼1 for the nucleotide composition of each triplet (Figs. 2B, 2D and Table S4). The two last codon positions have an elevated pyrimidine content (>50%), which is opposite to that of the 1st position of a codon; thus, again the Pur/Pyr content at the 2nd and 3rd positions compensates the strong bias from intrastrand DNA symmetry of the nucleotide composition at the 1st position of a codon in human CDSs. It should be noted that the increased A/T ratio at the 2nd position of a codon is compensated by a very low A content at the 3rd position (see N1 in CDS counts for the strand-independent case in Table S4). It is noteworthy that the statistical patterns we observed in the distribution of nucleotides within CDSs of bacteria and humans are universal and do not depend on the organism (Fig. 2). By contrast, codon usage bias (see the P1 column in the N3 section in CDS counts for the strand-independent case in Table S4), which refers to differences in the frequency of occurrence of synonymous codons, is not universal and varies strongly between different organisms.

Of note, due to redundancy of the genetic code, mutations at the 3rd position of a codon often result in a synonymous change of amino acid sequence of a protein, thereby explaining lower evolutionary conservation of this position as compared to the first two positions of a codon (Bofkin & Goldman, 2007). Mutations at the 1st and 2nd positions of a codon in the majority of cases lead to nonsynonymous changes in the protein sequences; consequently, it is widely thought that these mutations are subject to strong selection pressure. Therefore, a common view is that nucleotide compositions of the first two positions of a codon are much more functionally constrained, as compared to the 3rd one. Nevertheless, our data indicate that variations of nucleotide composition at the 3rd position of a codon exhibit remarkable statistical entanglement or interdependence with that of the 1st and 2nd positions, owing to the law of large numbers, which maintains CSPR at local and genome levels. On the basis of these observations, we believe that the functional constraints and selective pressure toward the first two codon positions also should act on the 3rd position of a codon, owing to the statistical entanglement of their nucleotide compositions. According to our hypothesis, an increase in the purine content at the 3rd position of codons, possibly owing to the synonymous transversion, is under strong selection pressure. Indeed, the 3rd position of the codon has much higher transition/transversion mutation bias than the 1st and 2nd positions do, suggesting that transversions are disproportionally selected against at the last position of a codon (Bofkin & Goldman, 2007). Thus, under “intrastrand DNA symmetry equilibrium” of the nucleotide composition of a gene, both synonymous and nonsynonymous mutations in the CDS should be under selective pressure because of their hidden contactless interactions, which we refer to as “statistical entanglement” here. In agreement with our observation, a recent study showed that 75% of synonymous mutations in 8,541 yeast strains resulted in a significant reduction in fitness, indicating strong non-neutrality of most mutations at the 3rd position of a codon (Shen et al., 2022).

Taken together, these observations suggest that the synonymous mutations at the 3rd position of a codon—when they accumulate in a sufficient number—are neither neutral nor nearly neutral but subject to natural selection to an extent similar to that of nonsynonymous mutations in the 1st and 2nd codon positions. Consequently, we propose that the ratio of nonsynonymous to synonymous substitutions (dN/dS) cannot be used as a measure of the strength and mode of natural selection (Shen et al., 2022), especially when acting on the protein products of genes involved in complex traits and pleiotropy. Hence, by virtue of CSPR, random mutations that occur in cellular genomes at various genetic loci and accumulate over generations, all would become statistically interwoven, and this phenomenon would make them all subject to natural selection despite the absence of identifiable functional mechanisms. Moreover, regarding the genetic loci, such as junk DNA in higher eukaryotes (Orgel & Crick, 1980), that are seemingly useless and have no known biological function, their nucleotide compositions are statistically interwoven with that of functional regulatory and coding DNA sequences; as a consequence, natural selection can act on the genetic variations in both functional DNA and junk DNA. Owing to the statistical entanglement, accumulation of mutations in junk DNA will exert increasing selective pressure on the nucleotide compositions and patterns of mutations in functional (regulatory and coding) DNA sequences in a contactless implicit manner, thus making natural selection to act on seemingly functionless junk DNA.

The standard genetic code and protein compositions comply with intrastrand DNA symmetry

At a local scale, nucleotide compositions of CDSs do not follow CSPR but rather obey Szybalski’s rule. These local CDS biases are due to the directional mutation pressure and intrinsic asymmetry in DNA transcription and replication (Lobry & Lobry, 1999; Sueoka, 1995). It is well established that orientations of genes and their CDSs on a chromosome are arranged so that the number of genes transcribed from the Watson (+) strand is nearly equal to the number of genes transcribed in the opposite direction, i.e., from the Crick (–) strand. This compensatory intermingling of CDS fragments balances the compositional biases of the protein-coding DNA sequences at the whole-genome level (Rapoport & Trifonov, 2013). It should be noted that an asymmetric distribution of coding sequences between DNA strands could be explained by GC/TA skews. Therefore, to analyze the nucleotide compositions at different positions of a codon, we performed an analysis of CDSs in humans and mice by taking into account only a single reference (+) Watson strand (see Watson (+) strand counts in Table S4). For this purpose, we converted CDSs from the antisense (−) Crick strand to their reverse complimentary analogs in which we kept 1st, 2nd, and 3rd codon position semantics but took a reverse complementary sequence with the orientation of the Watson strand. The occurrence of each of the four mononucleotides (A, C, G, and T) at the 1st, 2nd, and 3rd position of a codon was measured, and then A/T and G/C contents were calculated for each codon position and expressed as percentages (see N1 in CDS counts for the strand-dependent case in Table S4). The same measurements were performed for each of 16 dinucleotides and 64 trinucleotides, and relative differences between reverse complementary pairs (for example —CA—/—TG— and —TGC—/—GCA— ratios) were calculated (see N2 and N3 in CDS counts for the strand-dependent case in Table S4). The analysis of these data confirmed our observation made above that the nucleotide composition of the 3rd position of a codon compensates purine and pyrimidine biases of the 1st and 2nd positions (see N1 in CDS counts for the strand-dependent case in Table S4).

As depicted in Fig. 3, when taking into consideration the CDS shuffling between the Crick and Watson strands, we can reveal hidden patterns. By tracing over the lines to connect amino acids presented as squares, together with their respective codons and anticodons, it is possible to assemble amino acids into three clusters or groups, each creating a geometric shape: a triangle (Group I, α), a pentagram (Group II, τ), and a rhombus (Group III, β).

Figure 3 Demonstration of intrastrand DNA symmetry in the standard genetic code.

Instances of each coding triplet and of its reverse complementary counterpart in the reference DNA strand were counted and compared to each other. The relative difference between the number of instances of a triplet and that of its reverse complementary counterpart is expressed as a percentage. The amino acids and their codons are assembled into reverse complementary pairs connected by a solid line to create different geometrical shapes. On average, group I contains amino acids most commonly found in an α-helix, group II contains amino acids most commonly found in reverse turns (τ), and group III contains amino acids most commonly found in an extended structure (β-sheet) (Zull & Smith, 1990). See the text and Table S3 for details.

As expected, the number of codons was nearly equal to the number of their anticodons in a single DNA strand with differences ≥1% (Fig. 3 and CDS counts for the strand-dependent case in Table S4). Interestingly, these geometrical shapes have been constructed previously using reverse complementary interactions between codons and anticodons including stop codons (Heal et al., 2002; Mekler & Idlis, 1981). Our study provides a quantitative analysis of the reverse complementary interactions between amino acids and their codons and reveals that interactions between complementary amino acids and between sense and antisense peptides are governed by intrastrand DNA symmetry of protein-coding sequences. Therefore, we may say that the standard universal genetic code is a product of natural selection, which took place among many other previously existing alternative versions of the code, and this process led to codon assignments in compliance with CSPR. In other words, CSPR is one of the optimizing criteria for the selection of actual genetic codes among ancient alternative versions. Intriguingly, the pentagram encompasses eight amino acids plus a stop codon, whereas the triangle contains five amino acids plus two stop codons, and the rhombus contains seven amino acids and no stop codon. It did not escape our attention that (i) each amino acid cluster contains at least one aromatic, one basic, one polar, and one nonpolar residue; and (ii) each cluster that is composed of 8 + 1, 5 + 2, and seven amino acids plus stop codons can be encrypted by only two-letter genetic code, pointing to a possible origin of the standard genetic code.

Intrastrand DNA symmetry and the origin of additive genetic interactions

When a sufficiently big number of random mutations is taken into consideration, all mutations can be assembled into pairs of reverse complementary matches (see Table 1), indicating that CSPR restrains the number of possible DNA sequence variations. Nevertheless, a small number of random mutations may not always exhibit the expected intrastrand DNA symmetry, but with each additional mutation, the probability that all observed mutations are nearly consistent with CSPR increases. Consequently, when the number of mutations occurring in various genetic loci increases in a time-dependent manner, they would approach intrastrand DNA symmetry and exhibit strong statistical entanglement. For this reason, an increasing number of mutations in noncoding DNA and in peripheral gene networks should be compensated in part by the reverse complementary mutations in coding DNA and in the core gene network, respectively, and vice versa, so that the nucleotide compositions of an entire chromosome comply with CSPR. Furthermore, the homogenizing force of CSPR acts also at the local gene level; over the course of multiple generations, nucleotide composition of a CDS should closely approach intrastrand DNA symmetry while reconciling with the structural and functional constraints applied to the evolution of a protein sequence (Fig. 2, black squares).

The analysis of reverse complementary pairs of SNPs and entangled codon positions in the previous sections further demonstrated that intrastrand DNA symmetry is a general property of cellular organisms that underlies statistical interdependence of nucleotide compositions of different genetic loci and patterns of spontaneous mutations. Hence, we believe that this statistical entanglement represents a long-sought mechanism of the additive genetic interactions that constitute the basis for continuous variation of expression of complex traits. According to Fischer’s infinitesimal model, continuous phenotypic variation of a quantitative trait is implemented through additive genetic interactions of an infinitely large number of genetic loci (Fisher, 1919). Here, we propose that the additive genetic effect results from intrastrand complementarity of nucleotide compositions of diverse genetic loci participating in a trait variation. The nucleotide compositions of coding DNA sequences and regulatory elements (regulatory sequences that control gene expression, scaffold attachment regions, origins of DNA replication, centromeres, and telomeres) obey CSPR. In higher eukaryotes, these highly conserved DNA sequences are statistically interwoven with other sites in a genome, most of which are likely noncoding and nonconserved. Consequently, when a sufficient number of the genetic variants is accumulated in noncoding DNA sequences, this situation may cause continuous variation of a trait via modulation of the expression of highly conserved DNA sequences.

Because of CSPR, which is an example of the law of large numbers, random genetic mutations in a cell or an organism have a statistical tendency to accumulate in a pairwise manner, where a pair of reverse complementary SNPs in a population is present in near-equal numbers —X↔Y— ≈ —X’↔Y’— or —NXN↔NYN— ≈ —N’X’N’↔N’Y’N’— (e.g., the numbers of C↔T or ACT↔ATT SNPs in a population are near-equal to that of G↔A or AGT↔AAT SNPs, respectively [—C↔T— ≈—G↔A— or —ACT↔ATT— ≈ —AGT↔AAT—]). Consequently, according to CSPR, accumulation of random C→T mutations should be accompanied with a nearly exact number of random G→A mutations in the same DNA strand of a chromosome that are distributed arbitrarily anywhere in a genome. As illustrated in Fig. 4, we propose that accumulation of reverse complementary single-base substitution (SBS) mutations results in an additive genetic effect, which under directional selection could shift the expression of a phenotype toward one or the other side of the trait spectrum, because of the statistical entanglement of the highly conserved functional genetic loci such CDSs and regulatory elements (also referred to as “core” genes) with other loci (mainly noncoding and nonconserved DNA sequences) in a genome. For example, pairwise accumulation of either “forward” X→Y/X’→Y’ mutations (e.g., T→C/A→G) or “reverse” Y→X/Y’→X’ mutations (e.g., C→T/G →A) in a population of organisms or somatic cells would have an increasing additive effect on the expression of a trait (Figs. 4A, 4B). Importantly, mutations with small effect size should occur more likely at noncoding and nonconserved loci, whereas mutations with large effect size should occur in the highly conserved loci or “core” genes (Fig. 4A). According to our model shown in Fig. 1, the latter mutations may be under strong negative selection because of the large size of their effect on the expression of a trait (O’Connor et al., 2019). Figure 4B shows a hypothetical example of the statistical entanglement that is transformed into additive genetic interactions: accumulation of “forward” T →C/A →G mutations may shift a trait variation to the left or “minus” side of the normal distribution (e.g., toward a lower glucose consumption rate of a somatic cell), whereas “reverse” C→T/G→A mutations should shift the trait variation in the opposite direction to the right or “plus” side (e.g., toward a higher glucose consumption rate). It should be stressed that in most cases, each reverse complementary SBS is expected to have rather small effect size, but their contributions to a trait variation should be additive as long as they are in compliance with CSPR. Therefore, under conditions of directional selection, these SBSs can accumulate and yield a large effect size depending on the number of reverse complementary mutations. We hypothesize that the effect size of a mutation in the genome is inversely proportional to the degree of sequence conservation at a given locus (Fig. 4A) and to the distance between the SBS and the “core” gene(s) (Fig. 4C).

Figure 4 Graphical representation of the additive effect of random mutations on phenotypic variations of a quantitative trait.

Accumulation of the statistically entangled mutations owing to intrastrand DNA symmetry creates continuous trait variation. The majority of pairwise single-base substitution (SBS) mutations makes an infinitely small additive contribution to the trait variations (e.g., glucose uptake by a somatic cell). Accumulation of “forward” X→Y/X’→Y’ (T→C/A→G) mutations leads to an increase, whereas “reverse” Y→X/Y’→X’ (C→T/G→A) mutations lead to a decrease in expression of the trait. Under directional selection, a difference between the numbers of “forward” and “reverse” mutations increases. (A) The general case for the effect of SBSs on the normal distribution of a quantitative trait with a mean value of a trait (x¯) at the center of the X-axis. Note that the degree of DNA sequence conservation is indicated on an additional X-axis at the bottom. (B) A special case for the effect of defined SBSs on the normal distribution of a quantitative trait. (C) The normal distribution of a quantitative trait with the mean value of a trait (x¯) is at two extremes of the X-axis. Note that the distance from a core gene is indicated on an additional X-axis at the bottom.

According to this model, accumulation of random mutations in the pairwise manner [“forward” X→Y/X’→Y’ (e.g., T→C/A →G) and/or “reverse” Y→X/Y’→X’ (e.g., C→T/G→A)] over time and through multiple generations will give rise to continuous variation of the expression of complex traits in a population and at the same time will maintain intrastrand DNA symmetry. Under the directional selection that favors proliferation of a species or somatic cells with either decreased (−) or increased (+) expression of the trait, the accumulation of the reverse complementary mutations such as C→T/G→A or A→C/T→G with the additive effect may change the GC content, which is one of integral features of a genome. For example, the GC content would decrease, if due to differential selection, the “forward” C→T/G→A and C→A/G→T mutations accumulate at a higher rate as compared to the “reverse” T→C/A→G and A→C/T→G mutations, respectively. Besides, all types of SBSs including transversions C→G/G→C and T→A/A→T may increase or decrease local deviation from CSPR in GC/TA skews, respectively, if they accumulate in a chromosome at opposite positions with respect to the minimum or maximum of a skew.

The analysis of the patterns of common variants affecting human height, a classic example of a quantitative trait, revealed that the majority of 697 SNPs identified by GWAS are T*↔C (231) and A*↔G (245) (the asterisk denotes an effect allele) (Wood et al., 2014). Note that —T*↔C— ≈ —A*↔G— as expected, and surprisingly, no “reverse” C*↔T and G*↔A variants were found among the 697 variants. We suppose that these SNPs are statistically entangled with nucleotide compositions of some highly conserved “core” genes and loci, and the mutations occurring in these “core” elements may have large size of the effect on human height. Other examples of the additive effects of reverse complementary mutations are mutational signatures in cancer; in a recent version of the COSMIC (the Catalogue Of Somatic Mutations In Cancer) database v3.3 (June 2022), for SBSs, approximately 60 different types of mutational signatures are listed (Alexandrov et al., 2020). Remarkably, the majority of SBS signatures in cancer (48 out of 60) lead to a change in the GC content: for example, in SBSs 1, 2, 6, 7ab, 11, 15, 19, 23, 31, and 32, the number of C→T mutations is much greater than that of reverse T→C mutations; therefore, accumulation of the former causes a very small decrease in the GC content of cancer genomes. It should be pointed out that the algorithm that generates SBS signatures takes in account only six possible base substitutions instead of 12; this operation is performed by adding together pyrimidine and purine substitutions, e.g., C→T and G→A mutations are presented as C→T only, whereas C→A and G→T mutations are shown as C→A only; this algorithm does not tell us whether —C→T— ≈ —G→A—. In summary, accumulation of the reverse complementary T*↔C/A*↔G SNPs in the human population and of C→T/G→A mutations in certain common cancer types supports our hypothesis that these genetic variants are additive and shift the trait variation to either end of a normal distribution thereby providing a raw material (in the form of continuous variation) for natural selection (Fig. 4).

Polygenic adaptation and the transfer of genetic information

The gene-centered view of evolution (gene selection theory, i.e., selfish gene theory) postulates that natural evolution is mediated by differential survival of competing genes, which manifests itself as increasing frequencies of alleles whose phenotypes have higher fitness thereby enabling their propagation in a population (Dawkins, 1976). Francis Crick’s central dogma provides a molecular mechanism to the gene-centered concept of evolution by postulating the unidirectional strictly linear transfer of genetic information from DNA to proteins and phenotypes (Fig. 5). However, already in 1957, Haldane pointed out that natural selection cannot act simultaneously on multiple genes because of very high cost of selection (Haldane, 1957). For example, if natural selection acts on 10 independent genes coding for different traits, then among offspring, only one individual out of 1,024 may survive. As we mentioned above, mutations in coding sequences or functionally important regions have large effect sizes and are subject to strong negative selection, thus providing the molecular mechanism that underlies Haldane’s dilemma (O’Connor et al., 2019). To reconcile the observed higher-than-expected genetic polymorphism in natural populations with the cost of selection, Kimura proposed that a large proportion of the observed genetic variations must be selectively neutral and be retained via random drift (Kimura, 1968). On the other hand, since the introduction of Fisher’s infinitesimal model, geneticists have long agreed that adaptive variation is highly polygenic, and that this is necessary for efficient selection. Nevertheless, a central problem of genetic studies on quantitative traits is that most of the SNP variations found by GWASs are associated with only a small effect size and have very low predictive value. The omnigenic model proposed by Boyle, Li & Pritchard (2017) to explain the genetics of quantitative traits postulates that gene regulatory networks are so interconnected that the sum of small variations in the expression of thousands of individual genes in relevant cells may contribute to the trait. Although GWASs have successfully identified common variants contributing to polygenic phenotypes and diseases, the task of uncovering the causative molecular mechanisms via which significantly associated variants contribute to phenotypic variation is getting exponentially more and more complicated as the research progresses. It is now becoming evident that Crick’s central dogma’s mechanistic residue-by-residue approach fails to explain how hundreds of thousands of SNPs that are distributed uniformly across the genome contribute to phenotypic variation of a complex trait; this is because a gene’s function becomes diffuse and dependent on many distant separate loci. Molecules of DNA encode instructions that tell cells what to do; therefore, the identification of the molecular mechanisms of additive genetic interactions in Fisher’s infinitesimal model may not be possible within the existing gene-centric paradigm.

Figure 5 Schematic representation of the explicit and implicit modes of transfer of genetic information.

The central part of the picture represents the explicit mode or Crick’s central dogma, where arrows with a solid line represent linear residue-by-residue transfer of genetic information from DNA to a protein and phenotype (human height). At the periphery of the picture, arrows with a wavy dashed line represent the implicit mode of transfer of genetic information from DNA to a phenotype (human height). The silhouettes of DNA, RNA, protein and compound were prepared using Accelrys Discovery Studio 3.0 Visualizer software and open sources on the 3D structures of molecules.

Mechanistic models describe specific biologically relevant functions and deal with a finite number of genes and other genetic elements, which interact in an explicit manner to perform their functions and are encoded by a finite number of genetic loci. By contrast, complex traits, such as the size of cells, human height, and cancer, involve a large number of biological functions and are associated with their proteins and other macromolecules and consequently are encoded and regulated by a large number of genetic loci. Therefore, in genetics, the use of isogenic strains is a standard practice, which rules out an effect of a genetic background or of differences between the genomes of organisms under study. Even subtle differences in distant genetic loci may influence the expression of a gene or mutation under study owing to the genetic interactions. Thus, this trivial observation indicates that various types of genetic interactions can organize thousands of cellular genes into a highly interconnected network in which a random mutational process ultimately produces continuous phenotypic variation in the expression of a trait. We believe that the nature of additive genetic interactions in Fisher’s infinitesimal model cannot be understood in a pure mechanistic manner but rather by means of a statistical approach.

In the present work, we demonstrate that genetic polymorphism in human and murine populations exhibits intrastrand DNA symmetry that makes nucleotide compositions of different genetic loci statistically interwoven or entangled. By analyzing nucleotide composition of bacterial, human, and mouse CDSs at different positions of a codon, we demonstrate that the 3rd nucleotide in a codon is statistically interlinked with the 1st and 2nd nucleotides. As a consequence, natural selection can act on both synonymous and nonsynonymous mutations (Shen et al., 2022). Furthermore, owing to CSPR, the genetic variations in both coding and noncoding DNA loci can contribute to phenotypic variations and make natural selection act on seemingly functionless junk DNA. Here, we propose that CSPR is the symmetry of Fisher’s infinitesimal model and a fundamental feature of quantitative traits, which is conserved throughout evolution. Under this intrinsic DNA symmetry, the interactions between genetic loci occur rather in an implicit hidden manner and are based on the statistical relation between their nucleotide compositions, which we refer to as a statistical entanglement.

In the present study, we conclude that the statistical entanglement between the nucleotide compositions of different genetic loci provides a contactless implicit means for the transfer of genetic information from genome to phenotype (Fig. 5). In our version of Crick’s dogma, the statistical interdependence between different genetic loci makes additive genetic interactions in Fisher’s infinitesimal model possible and explains the diffuse function of a gene. Accordingly, nucleotide composition of a gene directly affecting a trait via physical action of the encoded protein is statistically interwoven with many different loci (in a genome), and each of which makes an infinitely small additive contribution to the phenotypic variations. Thus, the statistical entanglement of nucleotides in DNA challenges the neutral theory of molecular evolution. For the genomes complying with CSPR, there are neither neutral nor nearly neutral mutations, because of the additive genetic interactions in which all random mutations participate in a messengerless manner. Consequently, a cellular genome’s intrastrand DNA symmetry equilibrium, which originates from stochastic processes, generates a regulatory highly interconnected genetic network in living organisms.

Such features as intrastrand DNA symmetry, the GC content, complex polygenic traits, and genome architecture, which involve GC/TA skews, can be regarded as integral characteristics of a genome because they are dependent on a large number of genetic loci scattered across the entire genome. Here we propose that integral characteristics of a genome can be inferred as statistical interdependence between nucleotide compositions of multiple genetic loci. For example, GC/TA skews are localized deviations from CSPR and play functional roles in gene expression and DNA replication. Because of the compensatory nature of CSPR, large DNA sequence segments with opposite signs (e.g., G > C and G < C) are intermingled thereby establishing intrastrand symmetry (G ≈ C) over a sufficiently long DNA swath (Rapoport & Trifonov, 2013). This means that GC/TA skews enable the entanglement of nucleotide compositions of very distant genetic loci. We propose that these long-distance genetic interactions mediated by localized deviations from CSPR are essential for differential expression of genes and for embryonic development, and these in turn create selection pressure maintaining GC/TA skews and other features of genome architecture despite random genetic drift. From these observations, we can infer that morphology, as a complex polygenic trait in higher plants and animals, is the result of long-range genetic interactions caused by the compensatory role of CSPR. Therefore, the integral features of a genome such as genome architecture and quantitative traits are a cumulative result of multiple genetic alterations that have occurred under directional selective pressure during evolution.

Genomic sequences can be classified into two categories: highly conserved ones (such as ribosomal DNA, CDSs, and regulatory elements) and weakly conserved ones (such as noncoding junk DNA with no obvious function). Of note, evolutionary changes in discrete genomic elements such as protein sequence and ribosomal RNA occur steadily and comply with the molecular clock model (Zuckerkandl & Pauling, 1962). On the other hand, clock-like genetic alterations may not be suitable for describing the evolution of integral characteristics such as genome architecture (Koonin, 2009) and complex traits. Paradoxically, changes in highly conserved regions of a genome are unlikely to lead to dramatic alterations of integral characteristics of a genome but rather may be under strong negative selection. Given that the bulk of noncoding DNA in higher eukaryotes plays a more important role in local and global genome architecture, as compared to coding DNA, we could say that the resulting genomic design, as an ensemble of the localized deviations from CSPR, makes long-range genetic interactions possible via the whole-genome statistical entanglement. As a consequence, genetic variations in noncoding junk DNA may participate in the continuous variation of complex traits via long-distance communication with core genes. Furthermore, accumulation of mutations in noncoding junk DNA under strong directional selection may change genome architecture and other integral features, without dramatic alterations of the conserved DNA sequences. It is tempting to speculate that dramatic changes in the integral characteristics of a genome without accompanying changes in conserved genes can explain the phenomenon of evolutionary radiation such as the Cambrian explosion.

One of important consequences of CSPR is the restrained number of possible combinations of a DNA sequence. For example, the total possible number of combinations of a DNA sequence N nucleotides in length is 4N; however, under conditions G = C (S) and A = T (W), the number of DNA sequences perfectly complying with CSPR will be CNN22, which can be asymptotically reduced down to 4N/N. As N goes to infinity, the proportion of perfectly CSPR-compliant sequences tends to approach zero. In general, CSPR has an approximate and imprecise nature and is an evolutionarily stable property of chromosomal DNA in all cellular organisms. Perfectly CSPR-compliant sequences form a lattice in the space of all DNA sequences, which attracts and forces actual sequences of chromosomes to be very similar to ideally intrastrand-symmetric DNA sequences. Consequently, the evolution of chromosomal DNA sequences is constrained by CSPR. In fact, CSPR constrains frequencies of nucleotide sequences, except for self-complementary even-length DNA sequences, such as CG or CATG. Frequencies of mono-/di-/tri-nucleotides vary within a finite volume of frequency space and are constrained by the sum of all sequences’ frequencies equal to 1.0. For example, four mononucleotide frequencies FA, FT, FG, and FC form a three-dimensional finite-volume hyperplane owing to the constraint FA + FT + FG + FC = 1.0. By contrast, under perfect intrastrand DNA symmetry, FA = FT and FG = FC, and evolution of single-nucleotide frequencies could be pictured as a random walk in two-dimensional (2D) space, which in turn has an interesting property. A theorem by George Polya postulates that the probability of a return to an initial position is exactly 1.0 (i.e., this event will always happen) for symmetric random walks in 1D and 2D lattices but is strictly less than 1.0 in all higher-dimensional spaces and always decreases with each additional dimension. Due to the imprecise nature of CSPR, the space of mononucleotide frequencies is not exactly but quasi-two-dimensional; also, a DNA sequence’s random variations, having different mutational rates (depending on a nucleotide context)—just as directional mutations like CpG depletion—cannot always be considered as a symmetric random walk. However, it is tempting to speculate that quasi-2D structure of mononucleotide frequencies in the absence of directional mutational pressure will prevent neutral genetic drift and preserve integral characteristics during evolution, via the return to a starting point. At the level of mononucleotide frequencies, the evolution of single-nucleotide characteristics, such as the GC content, may have an oscillating nature. However, di- and tri-nucleotide frequencies vary in much higher-dimensional space and have a much lower chance to return to their initial values during evolution.

Conclusions

The gene-centered view of evolution and Francis Crick’s central dogma postulate strictly linear residue-by-residue transfer of genetic information from DNA to a phenotype in living organisms. Nonetheless, this scientific paradigm has some difficulties with explaining the role of junk DNA and the link between random mutations and continuous variation of complex polygenic traits. Here, we propose that CSPR makes nucleotide compositions of separate genetic loci statistically interdependent or interwoven. This statistical entanglement results in a diffused function of a gene or, in other words, makes the transfer of genetic information from nucleic acids to proteins and phenotypes implicit and messengerless (Fig. 5). Natural selection acts on many traits at once; accordingly, evolution proceeds via accumulation of small-effect-size mutations distributed over an entire genome. The statistical entanglement of stochastic genetic variations is a way to store information despite random genetic drift. Under these conditions, natural selection enables a species to occupy new niches via slow continuous accumulation of beneficial mutations, and this principle makes Darwinian evolution of a complex life form possible.

Supplemental Information

Supplemental Information 1 Supplementary Tables

Click here for additional data file.

Supplemental Information 2 List of the URL addresses of sequenced bacterial genomes

List of the URL addresses of sequenced bacterial genomes.

Click here for additional data file.

The authors would like to thank C.N.R.S. France and Nazarbayev University Kazakhstan for their support of basic research. The authors would like to thank Alexander Ishchenko for his assistance in preparation of the figure and Nikolai A. Shevchuk for manuscript review. The English language was corrected and certified by Shevchuk Editing.

Additional Information and Declarations

Competing Interests

Author Contributions

Data Availability

The authors declare there are no competing interests.

Bakhyt T. Matkarimov conceived and designed the experiments, performed the experiments, analyzed the data, prepared figures and/or tables, contributed analysis tools, and approved the final draft.

Murat K. Saparbaev conceived and designed the experiments, performed the experiments, analyzed the data, prepared figures and/or tables, authored or reviewed drafts of the article, and approved the final draft.

The following information was supplied regarding data availability:

The raw data used in this article come from open sources NCBT dbSNP dataset build 142 and NCBI RefSeq (Supplemental Files).

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
