# Peer review of "Chargaff’s second parity rule lies at the origin of additive genetic interactions in quantitative traits to make omnigenic selection possible"

_PeerJ, doi:10.7717/peerj.16671_

## Round 0.1 · original submission · Major Revisions

Dear Drs. Matkarimov and Saparbaev:

Thanks for submitting your manuscript to PeerJ. I have now received three independent reviews of your work, and as you will see, one reviewer recommended rejection, while another suggested a major revision. I am affording you the option of revising your manuscript according to all three reviews but understand that your resubmission may be sent to at least one new reviewer for a fresh assessment (unless the reviewer recommending rejection is willing to re-review).

Please work to make your revision clearer in its presentation and content. The authors provide numerous examples where clarity is needed. The methods should also be clear, concise, and repeatable. Please ensure this, and make sure all relevant information and references are provided. Also, elaborate on the discussion of your findings, placing them within a broad and inclusive body of work by the field (though terminology should be carefully used to make your work interesting to a broader audience).

Please fix all of the identified grammatical issues.

Therefore, I am recommending that you revise your manuscript, accordingly, taking into account all of the issues raised by the reviewers.

I look forward to seeing your revision, and thanks again for submitting your work to PeerJ.

Good luck with your revision,

-joe

Reviewer 1 ·

Basic reporting

The paper of Matkarimov and Saparbaev is a very interesting and stimulating work. They clearly show the relevance of Chargaff's second parity role in different genomes and its connection with the infinitesimal model. I find the results solid and have only a few minor points that I would like the authors to comment on.

Experimental design

1. The introduction is clearly divided into two blocks, the first discussing the background on CSPR and then introducing the infinitesimal model and its limitations. However, I would expect a concluding short paragraph or sentence clarifying how the authors intend to elucidate the contribution of CSPR to the evolutionary mechanisms of highly polygenic features of diseases.
2. Please, include the version or date of NCBI dbSNP and RefSeq databases download or analysis.
3. In the case of bacterial genomes, I miss some information about which strains were analyzed. Some bacteria split apart about 3000 MA, thus the comparison of diverse bacteria proteomes with human is quite uneven in my opinion.
4. Also regarding bacterial genomes, the mobilome may constitute an important part of the genome and also contribute to many polygenic traits. However, it is not clear to me whether the authors used genome assemblies (with chromosome and plasmids) or just the chromosome. Also, hypothetical proteins are very common in GMEs, so I am not convinced to exclude them as this could lead to unnecessary bias.

Validity of the findings

Regarding the conclusions of the work, I wonder if the authors consider that CSPR may help to uncover possible roles of junk DNA.

Additional comments

No further comments

Reviewer 2 ·

Basic reporting

The manuscript delves into an intriguing area of research, shedding light on some potential new mechanisms underlying genetic variation. However, the paper would benefit from more context and background, especially for readers not intimately familiar with the specifics of this field.

Experimental design

Consider elaborating on why SNPs from sex chromosomes were excluded, as this could be a potential limitation.
Analysis of nucleotide composition:

It's good that databases and exclusion criteria are mentioned, but more context on why specific databases were chosen might be helpful.

Please elaborate on the significance of strand orientation in this analysis. Why is it essential to consider gene orientation relative to the reference strand?

Validity of the findings

The result section is too verbose. Most informations in this section should be in the discussion section.

Reviewer 3 ·

Basic reporting

The manuscript by Matkarimov and Saparbaev presents an analysis of DNA base and triplet counts in human and mouse genomes with a specific focus on the positional relations in CDS codons, and on the single nucleotide polymorphism. The manuscript puts forward a conclusion of Chargaff’s second parity rule acting as an optimising force defining the observed sequence patterns at both contexts. The manuscript, though well written, and presenting a factually valid analyses and results in terms of the counts and observed symmetries in them, draws, in my opinion, a reverse and wrong conclusion (see below). With a number of additional confusing statements and the major stress on drawing of the major conclusion (starting from the title), I cannot recommend the work for the publication in its current state. Please, note however, that I find the dry results and the analyses valid, hence usable for reformulating the whole paper as a confirmatory analysis of the SNPs and codon biases also complying with Chargaff’s second parity rule.

Experimental design

Major concerns

A) The paper brings forward Chargaff’s second parity rule (PR2) as a constraint that shapes the observed mutational patterns (304-383), whereas the mutational biases (context dependent mutation rates) constitute a lower layer molecular characteristics that actually shape the higher level genomic properties, or a wider span sequence compositions, hence resulting in the PR2 compliance. In fact, since early years (see the works of Sueoka and Lobry, 1995), it was shown that any types of mutations can result in PR2 if the mutation rates for the same x->y conversion is the same at both strands. Some inconsistencies were present in the observed cases of strand variability of mutation rates, but those were shown to be still explainable by the mutation rate interrelations (see Pflughaupt 2023). The interrelations in mutations rates shape both the general patterns in codon compositions, and the SNP patterns, and the PR-2. Meaning PR-2 is a manifestation of the molecular level characteristics (mutation rates), just like the other two (SNP, codon) observations are, rather than PR-2 being a driving force in defining those two.

B) 263-302 describe how the symmetries in mutations may emerge as a result of the strand invariance of mutations propensity at both strands, however, to the reader it may seem as a novel result/interpretation rather than the initial proposal from Sueoka. Furthermore, that proposal was brought as part of a neutral evolution of the genomes, also invalidating the statement that PR2 is against the neutral theory of evolution (line 46). It would also help to discuss that the described mechanism is true only if the mutation rates are the same in both strands, because of which C->T = G->A, as G->A is a result of C->T mutation in the other strand.

C) 254-261 may be confusing for a reader to see “Medelian” used in the context of cancer, an inherently multigenic phenomenon hence the driver/passenger mutation definitions. I would suggest a reformulation and an inclusion of a “for an analogy” statement.

D) The discussion on the 1st, 2nd, and 3rd position entanglement in codons is not conclusive unless, again, matched with the patterns of mutation rates, in particular to the known heptameric sequence-context dependence of spontaneous mutation rates, as prior simulations with only such rates were shown to optimise a sequence to a state with expected triplet counts, hence may have nothing to do with any causality from PR-2.

Minor comments

a) In Materials and Methods, line 154, should not it be GCT <-> GAT instead of GAT <-> GCT?

b) In Abstract, line 27, there is an extra hypen in “sin-gle DNA strand”.

Validity of the findings

See above.

Additional comments

See above.

---

## Round 0.2 · accepted · Accept

Dear Drs. Matkarimov and Saparbaev:

Thanks for revising your manuscript based on the concerns raised by the reviewer. I now believe that your manuscript is suitable for publication. Congratulations! I look forward to seeing this work in print, and I anticipate it being an important resource for groups studying genetics and genome evolution. Thanks again for choosing PeerJ to publish such important work.

Best,

-joe

Reviewer 1 ·

Basic reporting

The revised version has gained in clarity of the research questions and approaches as well as the conclusions of the analysis. This clarity not only benefits the authors in terms of articulating their ideas but also greatly enhances the reader's understanding of the research.
In terms of the analysis and its conclusions, the revised manuscript has successfully strengthened the argument for a potential paradigm shift in the field. As the authors write, they have uncovered hidden patterns that should be further analyzed in the future. This revelation not only highlights the depth of their investigation but also points towards avenues for future exploration. In summary, the recognition of these concealed patterns opens up opportunities for further research, suggesting that this study is not just a standalone contribution but a catalyst for future inquiries.

Experimental design

No further comments

Validity of the findings

No comment

Additional comments

No further comments

Reviewer 2 ·

Basic reporting

The authors have addressed all my concerns.

Experimental design

The authors have addressed all my concerns.

Validity of the findings

The authors have addressed all my concerns.

Additional comments

The authors have addressed all my concerns.